# Mitochondrial Transfer to Host Cells from Ex Vivo Expanded Donor Hematopoietic Stem Cells

**DOI:** 10.3390/cells12111473

**Published:** 2023-05-25

**Authors:** Hiroki Kawano, Yuko Kawano, Chen Yu, Mark W. LaMere, Matthew J. McArthur, Michael W. Becker, Scott W. Ballinger, Satoshi Gojo, Roman A. Eliseev, Laura M. Calvi

**Affiliations:** 1Division of Hematology/Oncology, Department of Medicine, University of Rochester School of Medicine and Dentistry, Rochester, NY 14642, USA; 2James P. Wilmot Cancer Institute, University of Rochester School of Medicine and Dentistry, Rochester, NY 14642, USA; 3Division of Endocrinology and Metabolism, Department of Medicine, University of Rochester School of Medicine and Dentistry, Rochester, NY 14642, USA; 4Center for Musculoskeletal Research, University of Rochester School of Medicine and Dentistry, Rochester, NY 14642, USA; 5Division of Molecular and Cellular Pathology, University of Alabama at Birmingham, Birmingham, AL 35294, USA; 6Department of Regenerative Medicine, Kyoto Prefectural University of Medicine, Kyoto 602-0841, Japan

**Keywords:** mitochondrial DNA 2, ex vivo HSC expansion 3, MNX mouse, 5, in vivo mitochondrial transfer

## Abstract

Mitochondrial dysfunction is observed in various conditions, from metabolic syndromes to mitochondrial diseases. Moreover, mitochondrial DNA (mtDNA) transfer is an emerging mechanism that enables the restoration of mitochondrial function in damaged cells. Hence, developing a technology that facilitates the transfer of mtDNA can be a promising strategy for the treatment of these conditions. Here, we utilized an ex vivo culture of mouse hematopoietic stem cells (HSCs) and succeeded in expanding the HSCs efficiently. Upon transplantation, sufficient donor HSC engraftment was attained in-host. To assess the mitochondrial transfer via donor HSCs, we used mitochondrial-nuclear exchange (MNX) mice with nuclei from C57BL/6J and mitochondria from the C3H/HeN strain. Cells from MNX mice have C57BL/6J immunophenotype and C3H/HeN mtDNA, which is known to confer a higher stress resistance to mitochondria. Ex vivo expanded MNX HSCs were transplanted into irradiated C57BL/6J mice and the analyses were performed at six weeks post transplantation. We observed high engraftment of the donor cells in the bone marrow. We also found that HSCs from the MNX mice could transfer mtDNA to the host cells. This work highlights the utility of ex vivo expanded HSC to achieve the mitochondrial transfer from donor to host in the transplant setting.

## 1. Introduction

Mitochondria are highly dynamic organelles that regulate cell bioenergetics and fluxes of key metabolites and ions [1]. Their diverse functions are regulated by fission and fusion machinery, factors controlling biogenesis, ion transport, quality control mechanisms such as autophagy/mitophagy, and other factors [2]. Mitochondrial impairment has been shown to play a role in various pathophysiologic states, including organ dysfunctions due to degenerative changes, metabolic syndromes such as diabetes, aging, sustained chronic inflammation, space flight, and cancers [3,4,5]. There is also a range of mitochondrial diseases caused by mutations in the nuclear DNA- or mitochondrial DNA (mtDNA)-encoded mitochondrial genes, such as the Leigh syndrome and other conditions [6]. Various approaches have been used to elucidate the effects of mitochondria and mtDNA on biological processes, particularly multi-factorial events such as cancer and aging. Higher vulnerability of the mitochondrial genome to ethidium bromide relative to the nuclear genome has long been used to eliminate the mitochondrial genome, and cells that have been exposed to ethidium bromide and have lost their mitochondria are called ρ0 cells [7]. The result of fusion of an ρ0 cell and an enucleated cell is called a cybrid cell, and it has played an extremely important role in studying the involvement of the mitochondrial genome in the cancer field [8]. On the other hand, the use of cybrid cells is hampered by the fact that the effects of the carcinogenic properties of ethidium bromide on the nuclear genome cannot be ruled out. Taking advantage of the fact that the mitochondrial genome is maternally inherited, mice called conplastic mice have also elicited significant findings in aging and other areas [9]. To create these mice, the male lineage is fixed, female offspring born to the male lineage are used, and the female offspring are crossed with that male lineage again. By continuing this process, a new combination of mitochondrial and nuclear genomes can be created, with the mitochondrial genome derived from the female and the nuclear genome from the male [10]. On the other hand, using assisted reproductive technology, a new mitochondrial genome/nucleus genome combination can be created using nuclear transplantation, generating mitochondria nuclear exchange (MNX) mice [11]. A large time savings compared to conplastic mice has been achieved this way. A number of results have been reported using MNX mice, including that the mitochondrial genome alters the cancer microenvironment and the metastatic potential [12], that it alters immune cell differentiation [13], and that the molecular mechanism underlying their phenotypic change is the alteration of the nuclear epigenome through mitochondrial intermediate metabolites [14]. 

Recently, evidence has been accumulating that mitochondrial transfer occurs in different cell types, both in vitro and in vivo [15], in which donor cells, e.g., mesenchymal stem cells (MSCs), transfer mitochondria to other cells [16]. Many studies suggest that mitochondrial transfer can help the recovery process in injured cells or tissues [17]. MSCs have been evaluated as a practical source of mitochondrial transfer in the regenerative treatment for deteriorated kidneys in diabetes [18] or damaged cerebrovascular systems in strokes [19]. However, other primary cells, such as HSCs, have not been fully explored as a source of mitochondrial transfer. Hematopoietic stem cells (HSCs) are multipotent, self-renewing cells that regenerate the blood system after transplantation [20]. However, until recently, we lacked the technology to efficiently expand HSCs ex vivo until the development of a novel system where the culture media is supplemented with a unique replacement for serum albumin, polyvinyl alcohol (PVA) [21,22,23]. This novel technology can potentially enable us to manipulate primary HSCs ex vivo. 

In this work, we have thoroughly evaluated the ex vivo expansion of mouse HSCs from young or middle-aged mice. We present our data indicating a reproducible quality of the ex vivo culture of HSCs and that the culture from MNX mice shows the preserved functional HSCs, demonstrated utilizing the lethally irradiated transplant model [24]. We also identified mitochondrial transfer from donor HSCs to host cells by analyzing mtDNA in each population after transplant utilizing MNX mice, which possesses C57BL/6J (C57) nuclear and C3H/HeN (C3H) mitochondrial DNA [11]. These mice have the C57 immunophenotype, thus cells from the MNX mice can be safely transplanted into the C57 hosts without triggering any immune response. Notably, mitochondria in these MNX mice inherit the bioenergetic economy of the C3H strain when compared to the C57 strain [11]. It was previously shown that the donor C3H mtDNA haplotype promotes the survival of host cells as C3H mitochondria produce less ROS and are more stress-resistant [25,26]. We, therefore, hypothesized that the more resistant mitochondria of the donor MNX cells would have high chances of transfer into the less-resistant C57 host cells.

## 2. Materials and Methods

### 2.1. Study Approval

All murine studies were performed in accordance with protocols approved by the Institutional Animal Care and Use Committee, the University Committee on Animal Resources (University of Rochester, Rochester, NY, USA). Approval number is UCAR 2002-137E from 10 August 2022. This study was performed in accordance with ARRIVE guidelines.

### 2.2. Mice

Mice were maintained within the Vivarium facility at the University of Rochester School of Medicine and Dentistry in accordance with protocols approved by the Institutional Animal Care and Use Committee. C57BL/6J were purchased from The Jackson Laboratory and bred in-house. All strains were of the C57BL/6J background and expressed the CD45.2. Strains used include C57BL/6J, PepBoy CD45.1 (B6.SJL-Ptprc^a^PepC^b^/BoyJ), UBC-GFP (C57BL/6-Tg (UBC-GFP) 30Scha/J), and MNX (C57BL/6-MNX: C57n;C3Hmt) and originated in the laboratory of Dr Scott Ballinger (U of Alabama at Birmingham).

### 2.3. Complete Blood Counts 

Blood was collected from the submandibular plexus and collected in EDTA-coated tubes. The scil Vet abc Plus+ was used to analyze complete blood counts, as we previously published [27].

### 2.4. Light Microscopy 

Images were taken at room temperature using an Olympus CKX41 microscope and Olympus DP74 camera. Cellsens software (Olympus, Tokyo, Japan) was used to acquire images on the microscope.

### 2.5. Flow Cytometry

Analysis of marrow cell populations was performed as previously described [27]. For marrow cell analysis, marrow was released by crushing with a mortar and pestle in 1x PBS. Analysis for hematopoietic and mature cell populations was performed as previously described [28]. Samples were run on a BD LSR Fortessa flow cytometer: 5 lasers, UV (355 nm), violet (405 nm), blue (488 nm), yellow–green (561 nm), and red (640 nm) lasers (BD Biosciences, San Jose, CA, USA). As a dead stain, DAPI was used. Analysis was performed using FCS Express version 7 (De Novo Software, Pasadena, CA, USA). The gating strategy used to identify populations enriched for cells of interest has been previously described [22]. Sorting was performed on a FACSAriaⅡ with 405-, 488-, 532- and 640-lasers (BD Biosciences, San Jose, CA, USA).

### 2.6. HSC Isolation by Fluorescence-Activated Cell Sorting

Ex vivo mouse HSC collection was performed as previously described [21]. Briefly, mouse bone marrow cells were isolated from the tibia and femur and stained with APC-c-Kit antibody. C-Kit-positive cells were enriched using anti-APC magnetic beads with auto MACS pro (Miltenyi Biotec, Bergisch Gladbach, Germany). The c-Kit-enriched cells were subsequently stained with a lineage antibody cocktail (biotinylated CD3, TER119, LY-6G/LY-6C, and B220), anti-CD11b, anti-CD34, anti-Sca1, and anti-CD150 before being stained with streptavidin APC for 60 min. Antibodies used are listed in Appendix A. Cell populations were then purified using a FACS AriaⅡ and plated into the designated plate. Isolation and analyses were performed using serum-free PBS.

### 2.7. HSC Ex Vivo Culture 

Purified HSC cells were cultured in serum-free medium with PVA, HemEX-Type9A medium (Cell Science & Technology Institute, Miyagi, Japan, A5P00P01C) supplemented with 1% P/S (Gibco, Billings, MT, USA), 100 ng/mL mouse TPO (Peprotech, Rocky Hill, NJ, USA), 10 ng/mL mouse SCF (Peprotech, Rocky Hill, NJ, USA) at 37 °C with 5% CO_2_. Complete medium changes were performed every 2 days after the first 3 days as described [21]. All the cultures were performed utilizing flat-bottomed plates, tissue-culture-treated coated with fibronectin (Corning, Corning, NY, USA).

### 2.8. Quality Tests of Cultured Cells

Following ex vivo culture, cells were manually counted using hemocytometer. For flow cytometric analysis, cells were stained with antibodies described in Appendix A for 30 min. Following a wash step, the cells were run with a flow cytometry as mentioned above.

### 2.9. Transplant Assays

For non-competitive transplants, 5 × 10^5^ ex vivo expanded donor HSC cells were intravenously injected via the tail vein into recipient mice. Recipient mice were conditioned with a lethal dose of radiation using a ^137^Cs source (2 doses of 6 Gy total body irradiation, first at 24 h prior and the second at 1–3 h prior to transplantation).

### 2.10. Detection of C3H and C57 mtDNA

This was performed according to the previously published protocol [11] using specific PCR probes, PCR, and subsequent restriction digest. Briefly, viable donor (GFP−) and recipient (GFP+) cells were sorted and DNA extracted using the Wizard SV DNA kit (Promega, Madison, WI, USA). C3H mouse mtDNA carries a SNP in the *mtNd3* gene forming a *Bcl1* restriction site. To detect C3H mtDNA, we amplified the region carrying this restriction site using the following primers: 5′-TTC CAA TTA GTA GAT TCT GAA TAA ACC CAG AAG AGA GTG AT-3′ and 5′-AAA TTT TAT TGA GAA TGG TAG ACG-3′. PCR was performed and the product cut with *Bcl1*. The C3H mtDNA was cleaved into 166 bp and 38 bp fragments, while the C57 mtDNA remained uncut.

C57 mouse mtDNA carries a SNP in the *mtCo3* gene forming a *Pflf1* restriction site. To detect C57 mtDNA, we amplified the region carrying this restriction site using the following primers: 5′-CGA AAC CAC ATA AAT CAA GCC C-3′ and 5′-CTC TCT TCT GGG TTT ATT CAG A-3′. PCR was performed and the product cut with *Pflf1*. The C57 mtDNA was cleaved into 274 bp and 111 bp fragments while the C3H mtDNA remained uncut.

### 2.11. Statistical Analysis

All data are reported as mean ± standard error of the mean. All analyses were made with GraphPad Prism software (version 9) using 2-tailed Student’s *t* test, Mann–Whitney nonparametric testing, or 1-way ANOVA with Tukey’s multiple-comparisons post-test when appropriate. A *p* value less than 0.05 was considered significant and denoted by asterisks (* *p* < 0.05, ** *p* < 0.01, *** *p* < 0.001).

## 3. Results

### 3.1. Ex Vivo Culture of Mouse HSCs from Young Mice Reveals Robust Expansion Pattern and Reserved Functional HSCs in the Transplant Assay

To establish and confirm the reproducibility of ex vivo culture of mouse HSCs, we first harvested bone marrow from young mice and isolated HSCs according to the previously reported protocol [22]. The recent report also implies that even minor differences in culture constitutes can affect the maintenance of HSC in vitro [22,23]. In our experience, intense monitoring of cell morphology was required to time media changes. We used alternative culture media that are also described as optional by Wilkinson et al. [22] (see Methods). Moreover, we found that a stringent cKit+Sca1+lineage− (KSL)CD150+CD34- gate was needed to sort an HSC population that would expand ex vivo in these culture conditions and to avoid the contamination of hematopoietic progenitor cells. The percent of each gating population was almost similar to that previously described (Figure 1). The expansion of both the total and the KSL cell population was achieved very efficiently as assessed by flow cytometry (Figure 2a). The morphology of the cells in the culture was consistent with the previous report in which small cells were dominant with some large cells, thought to be megakaryocytes (arrowhead), present (Figure 2b). To evaluate the presence of functional HSCs, we performed conditional transplants utilizing lethal irradiation. We injected increasing doses of CD45.2BL6 HSCs into CD45.1 ubiquitin C/green fluorescent protein (UBC/GFP) recipient mice, which gave us the opportunity to evaluate the composition of the HSC/hematopoietic progenitor cell (HPC) or the bone marrow mature cells based on both CD45 subtype and GFP positivity (Figure 2c). The percent of donor cells in the peripheral white blood cells or bone marrow increased in a cell-number-dependent manner (Figure 2d). The complete blood counts (CBCs) of the recipient mice indicated hematopoietic reconstitution by the donor cells, which was also compatible with high donor engraftment in the bone marrow at 16 weeks (Figure 2e). For further evaluation of donor cell presence in the bone marrow, we analyzed whole bone marrow with flow cytometry based on the approach we previously described [27].

### 3.2. Ex Vivo Expanded Mouse HSCs Efficiently Replaced Host Hematopoietic Progenitor Cells (HPCs) or HSCs in a Donor Cell Number-Dependent Manner

The bone marrow cells were analyzed in the recipient mice 16 weeks after transplantation. The donor or recipient cells were clearly separated by flow cytometry based on the GFP expression. Each HPC/HSC population was dominated by donor-derived cells. It is known that increasing the number of donor stem cells can enhance the engraftment in BMT [29]. Thus, as we expected, the donor cell titration revealed the dose-dependent increases of donor engraftment (Figure 3a). Consistent with HSC differentiation, the composition of mature cells in the bone marrow also showed the predominance of donor-derived cells as was seen in the HPC/HSC compartment (Figure 3b). The chimerism of the donor cells was relatively low in T cells, which suggests the presence of residual persistent recipient T cells.

### 3.3. The Ex Vivo HSC Culture from MNX Mice Showed a Comparable Level of CD150+ HSCs with Potential of Engraftment after Transplantation

Next, we assessed the ex vivo HSC culture utilizing adult MNX mice (4 to 10 months old). As was described above, MNX mice have C57 nuclear genome and C3H mtDNA [11]. Since C3H and C57 mtDNA have different genetic markers easily identified via PCR and restriction digest [11], this system allows convenient detection of donor C3H mtDNA in GFP+ C57 host cells. As we expected, MNX mice showed a clear separation of KSL and CD150+ HSC populations (Figure 4, Gate 7 and Gate 8), similar with HSCs from young wild type donors (Figure 1, Gate 7 and Gate 8). Evaluation of the ex vivo culture after sorting the HSCs revealed a retention of CD150+KSL populations in the culture (Figure 5) with a growth pattern comparable to that seen in wild type HSCs (Figure 5). We then injected 5 × 10^5^ ex vivo expanded MNX HSCs into lethally irradiated CD45.1 UBC/GFP (C57 background) recipient mice (Figure 6a). Since we expected that mitochondrial transfer would occur at early time points after the transplantation, we analyzed the recipient mice at 6 weeks, which is an early time point when hematopoietic engraftment should have stabilized. Consistent with this, the CBC data revealed almost complete recovery by 6 weeks post transplantation (Figure 6c). Therefore, at six weeks post transplantation, we harvested bone marrow and the spleens for further evaluation of mitochondrial transfer from the MNX donor cells to the recipient cells. The chimerism in each tissue was very high, especially in the bone marrow (Figure 6b). In parallel with this, we found that the HPC/HSCs in the bone marrow were highly replaced with donor-derived cells, and each population cell number was comparable with the non-transplant, age- and sex-matched mice (Figure 7a), which implies robust reconstitution capacities of donor cells in vivo. Furthermore, the donor cell distribution in the mature bone marrow population was consistent with this (Figure 7b).

### 3.4. mtDNA Transfer from Donor to Host in a Transplant Model Utilizing Ex Vivo Expanded Mouse HSC Culture

To investigate the potential of the donor HSCs to transfer their mitochondria and, thus, mtDNA to the host, we extracted cells from the bone marrow and spleens of the recipient mice for mtDNA analysis. As described above, the recipient mice were of C57 background while the donor HSCs were from MNX (C57 nucleus and C3H mtDNA). The mt-Co3 gene of C57 mice carries a SNP that forms a Pflf1 restriction site and, thus, can be cleaved with the restriction enzyme Pflf1, while that of C3H cannot. On the other hand, the mt-Nd3 gene of C3H mice carries a SNP that forms a Bcl1 restriction site and, thus, can be cleaved by the restriction enzyme Bcl1, while that of C57 cannot. By observing the different bands generated by digesting the mtDNA with these restriction enzymes, we can trace the origin of the mitochondrial genome. We separated GFP− donor cells and GFP+ host cells via sorting. However, we could not attain the DNA analysis in GFP+ cells in the bone marrow due to the extremely low frequency of these cells. From the spleen, we received sufficient numbers of both donor (GFP−) and host (GFP+) cells. With a distinct separation of GFP− and GFP+ cells in the spleens, we have found that the donor C3H mtDNA was transferred to the GFP+ host cells (Figure 8a). In particular, after the PCR amplification of either mt-Co3 or mt-Nd3 gene fragments and digestion of these fragments with either Pflf1 or Bcl1, respectively, the host GFP+ cells showed three bands (one uncut wild type band and two cut products from SNP-carrying), suggesting that the donor (C3H) mitochondrial genome coexisted in the recipient cells with the recipient (C57) mitochondrial genome. The reverse transfer of mtDNA from recipient to donor was also detected but to a much lower extent, which suggests bi-directionality of mitochondrial transfer (Figure 8b).

## 4. Discussion

In the current study, we have demonstrated that ex vivo expansion of mouse HSCs can be attained with the careful monitoring and sorting approach gating KSL cell population. We also observed that donor-to-host mitochondrial transfer occurred in recipient mice transplanted with HSCs from this culture system. HSC culture ex vivo has been intensely explored due to the necessity of HSCs for basic research as well as in clinical aspects. This is despite the difficulties of maintaining HSCs while preventing cell differentiation into mature cells. Takubo et al. has developed an in vitro conditioning system to maintain the quiescent HSC in culture under low cytokines concentrations of stem cell factor (SCF) and thrombopoietin (TPO), hypoxia, and high amounts of fatty acid, with which they attained a 30–40% donor chimerism at four months after transplantation [30]. Wilkinson et al. have utilized PVA to replace serum albumin in mouse HSC in ex vivo culture to avoid the contamination of cytokines that induce differentiation of HSCs [21,22]. This approach has shown long-term stable expansion of functional HSCs ex vivo and sufficient engraftment capacity in vivo. Our culture, following the protocol by Wilkinson, showed relatively rapid HSC growth to reach subconfluency in 2 weeks, which is comparable to their result (8133× increase on day 28). However, very efficient engraftment or expected HSC numbers in vitro have been observed, which suggests that an ex vivo HSC expansion pattern can vary, possibly depending on multiple factors including the culture condition and gating strategy of the HSCs. Although the trafficking of exogenous mitochondria associated with their transfer has been reported through tunneling nanotube either in vitro [31] or in vivo [32] and trafficking vesicles, such as exosome [17] and exopher [33], the phenomenon of mitochondrial substitution has never been recognized in the context of mitochondrial transfer. Mitochondrial diseases, especially those caused by mutations in the mitochondrial genome, are still rare, intractable diseases for which there is no cure [34]. In recent years, significant progress in genome-editing technology has led to the proposal of methods to modify the genome by up to 50%, targeting mutations with point mutations [35]. Mitochondrial genome mutations do not develop phenotypic alternations until a certain threshold is crossed, and there are cases where mitochondrial genome mutations cross the threshold and phenotypically develop when a child is delivered from a mother of a carrier. Assisted reproductive technology has developed a mitochondrial replacement method that applies nuclear transfer to establish fertilization with the mother’s nucleus with third-party oocyte cytoplasm and the father’s sperm [36], and this has already reached clinical practice [37]. However, even with this elaborated technique, a re-dominance, or reversion, of the residual, slightly mutated mitochondrial genome of maternal origin has been observed [38]. The protocol that has made this mitochondrial genome replacement possible in somatic cells, whereby mitochondrial genomes are reduced once and then internalization of isolated mitochondria is promoted, has been reported to result in complete mitochondrial genome replacement in most cells [39]. Unlike these previous reports, in the current study, we demonstrate mitochondrial genome replacement in the setting of transplantation, where there is mixing of donor and recipient cells. Although it is completely unknown how heteroplasmy, which represents the heterogeneity of the mitochondrial genome, is regulated [40], methods to modulate heteroplasmy would greatly contribute to the development of treatment strategies for mitochondrial diseases for which currently no treatment methods exist. 

Some reports suggest that intercellular mitochondrial transfer occurs under both physiological and pathological conditions via donor cells, including MSC or astrocytes. This phenomenon potentially contributes to maintaining homeostasis or supporting recovery after tissue injury. Despite the limitation of attaining sufficient numbers of host cells in a transplant model, we were clearly able to detect the mitochondrial transfer from donor cells to host cells in the spleen utilizing ex vivo expansion culture systems, demonstrating the ability of HSCs and their progeny to donate mitochondria, even after ex vivo expansion.

Golan et al. have recently revealed that CD45+ donor cells contribute to mitochondrial transfer to bone marrow MSCs via cell–cell contact for the recovery from the irradiation-associated damages in conditional mouse transplants [41]. Interestingly, they also found opposite mitochondrial transfer from recipient to donor HSCs utilizing Dendra2-mitochondria transgenic mice [42], which is consistent with our finding utilizing highly purified HSC cells in ex vivo culture. Collectively, these findings suggest complex, bi-directional interactions between donor and recipient hematopoietic stem and progenitor cells (HSPCs) as well as bone marrow mitochondrial cells that occur to reconstitute hematopoietic system. While the transfer of mitochondria and, therefore, mitochondrial genomes following hematopoietic stem cell transplantation was previously reported [43], that exchange occurred in the bone marrow, where donor cells had been engrafted. In the present study, however, we found that progenies differentiated from transplanted stem cells could supply mitochondria to the recipient cells in the spleen, suggesting the possibility that mitochondrial trafficking can occur not only in the bone marrow but also in remote organs following hematopoietic stem cell transplantation. To our knowledge, this is the first report of this occurrence. Whether this transfer can also occur in other key organs, such as the liver and/or muscles, should be examined in future studies.

Mitochondria, involved not only in energy metabolism but also in epigenome-mediated regulation of cell differentiation [44] and intracellular innate immunity [45], are intercellularly more dynamic than ever thought possible. It would be of great impact to elucidate how mitochondria move between cells, what kind of cells perform this trafficking, and by what mechanism. Our current study opens the door for future investigations of these mechanisms. 

In summary, we have evaluated and confirmed robust expansion of mouse HSCs ex vivo. This culture system, while requiring careful monitoring and optimization of culture constituents, has the potential for a broad range of applications in HSC research. We have also established that using this technology enables us to demonstrate mitochondrial transfer between donor and recipient cells. Our results suggest that this can potentially be a novel approach for understanding the remodeling of the bone marrow microenvironment after transplantation and may represent a novel therapeutic strategy to mitigate not only mitochondrial diseases but also cancer and neurodegenerative diseases where mitochondrial genome and dynamics are disrupted.

## Figures and Tables

**Figure 1 cells-12-01473-f001:**
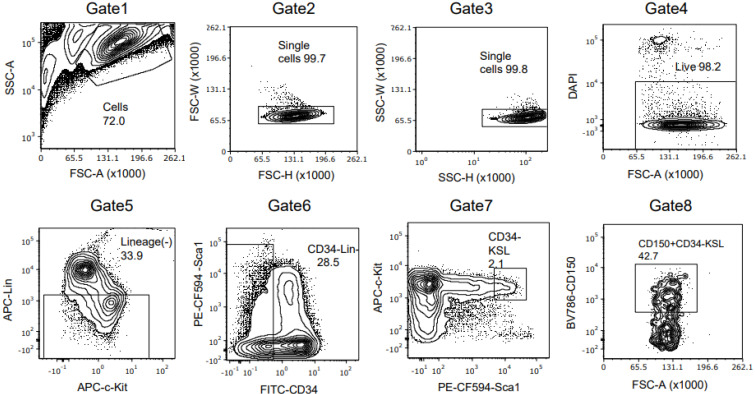
Representative gating scheme for FACS purification of mouse HSCs from young, wild type mice (from at least three independent experiments). Live CD150 (SLAM)^+^CD34^−^Kit^+^Sca1^+^Lineage^−^ HSCs were isolated by FACS from c-Kit-enriched bone marrow cells with the gating scheme and directly sorted in fresh HSC complete medium. FSC-A (forward scatter area), FSC-H (forward scatter height), FSC-W (forward scatter width), SSC-A (side scatter area), SSC-H (side scatter height), SSC-W (side scatter width).

**Figure 2 cells-12-01473-f002:**
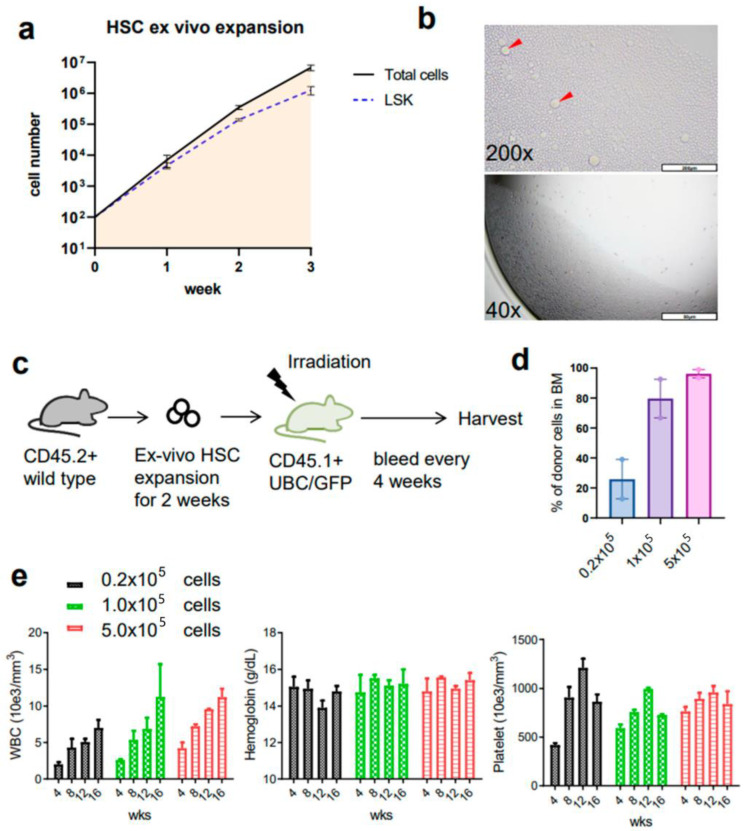
Robust ex vivo expansion of HSCs from young wild type mice. (**a**) The cell number of total cells and KSL (cKit+Sca1+Lineage−) during ex vivo culture. Total cell number was manually counted and KSL cell number was evaluated from flow cytometry. Data represent mean ± SEM from three independent experiments. (**b**) The representative images of ex vivo HSC culture. The cells expanded in 96-well, fibronectin-coated plate, and the images are expected cell density on day 11 at 40× and 200× magnification. Red arrows indicate larger (megakaryocyte-like) cells in the cultures. (**c**) Scheme of the noncompetitive transplant of ex vivo HSC donor cells into recipient mice with lethal irradiation. HSC cells were purified from CD45.2BL6 bone marrows and expanded in vitro for 2 weeks. Cells were injected into CD45.1 UBC/GFP mice after the 12 Gy of irradiation. The donor cell numbers were titrated in three groups (0.2 × 10^5^, 1 × 10^5^, and 5 × 10^5^ donor cells per recipient). The recipient mice were harvested in 16 weeks to analyze the bone marrow cells. (**d**) The chimerism of donor cells in PB at 4, 8, 12, and 16 weeks post-transplant, and in bone marrow at 16 weeks (n = 2 recipients per group). (**e**) CBCs of recipient mice. Mice were bled to assess CBCs every 4 weeks after transplant.

**Figure 3 cells-12-01473-f003:**
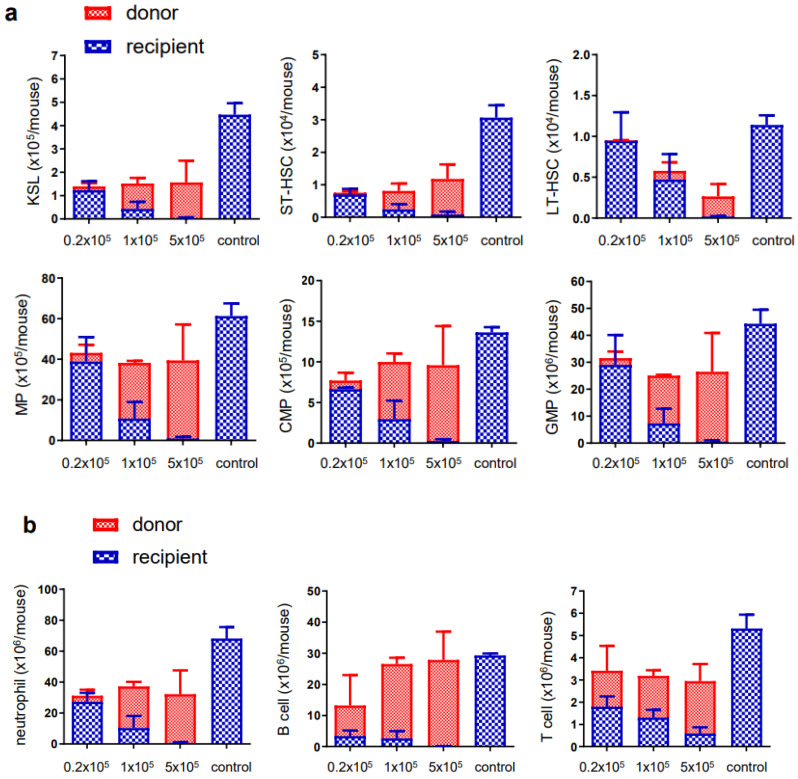
Donor cell dose-dependent contribution to bone marrow HSPCs for ex vivo expanded HSCs. Donor cell titration was performed (0.2 × 10^5^ cells, 1.0 × 10^5^ cells, and 5 × 10^5^ cells per shot for lethal irradiated recipients). (**a**) The cell number of donor or recipient cells in each HSPC population. (**b**) The mature cell number of donor or recipient cells. Control mice were not irradiated and were matched with age and sex of recipient mice. Red highlighted is donor and blue highlighted is recipient cell determined with flow cytometry. The values were displayed in a 2D stacked column graph. MP (myeloid progenitors) (Lin^−^/c-Kit^+^/Sca1^−^), CMP (common myeloid progenitors) (FcγR^−^/CD34^+^), GMP (granulocyte-macrophage progenitors) (FcγR^+^/CD34^+^), ST-HSC (short term-HSC) (KSL^+^CD150^−^CD48^−^Flt3^−)^, LT-HSC (long term-HSC) (KSL^+^CD150^+^CD48^−^Flt3^−^).

**Figure 4 cells-12-01473-f004:**
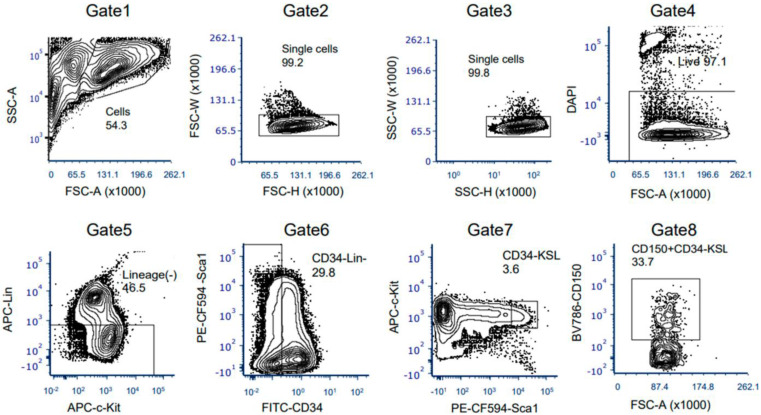
Gating scheme for FACS purification of bone marrow HSCs from MNX mice. Live CD150 (SLAM)^+^CD34^−^Kit^+^Sca1^+^Lineage^−^(KSL) HSCs were isolated by FACS from pooled bone marrow cells from MNX aged mice as described above (two independent experiments). HSCs were obtained from two donor mice for each experiment.

**Figure 5 cells-12-01473-f005:**
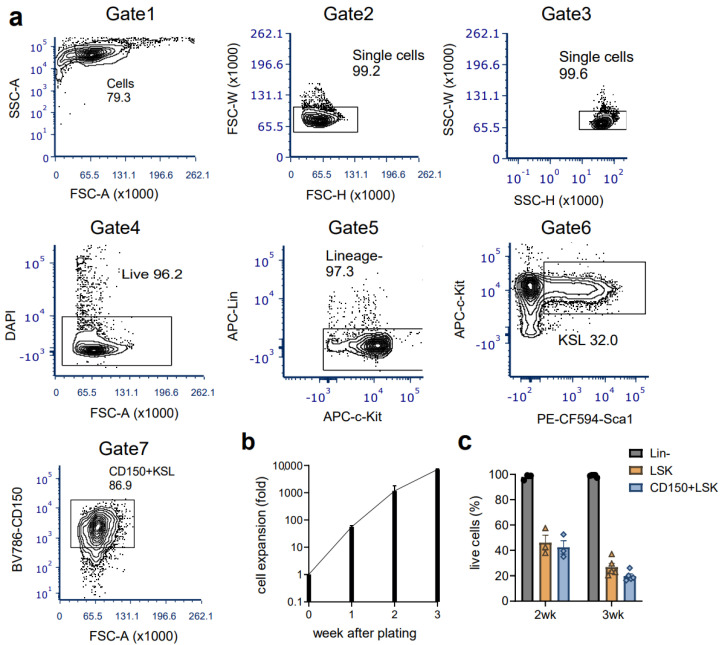
The representative gating scheme of quality check for ex vivo HSCs from MNX aged mice bone marrows. (**a**) HSC cultures were analyzed by flow cytometry using the gating scheme as shown on day 21 after sorting HSCs before the transplant (the representative gating plots from n = 5). (**b**) Cells expanded in vitro HSC culture were counted every week. (**c**) The quality check was regularly performed until the transplant (data from two independent experiments). Lin^−^ (lineage negative), KSL (cKit^+^Sca1^+^Lineage^−^).

**Figure 6 cells-12-01473-f006:**
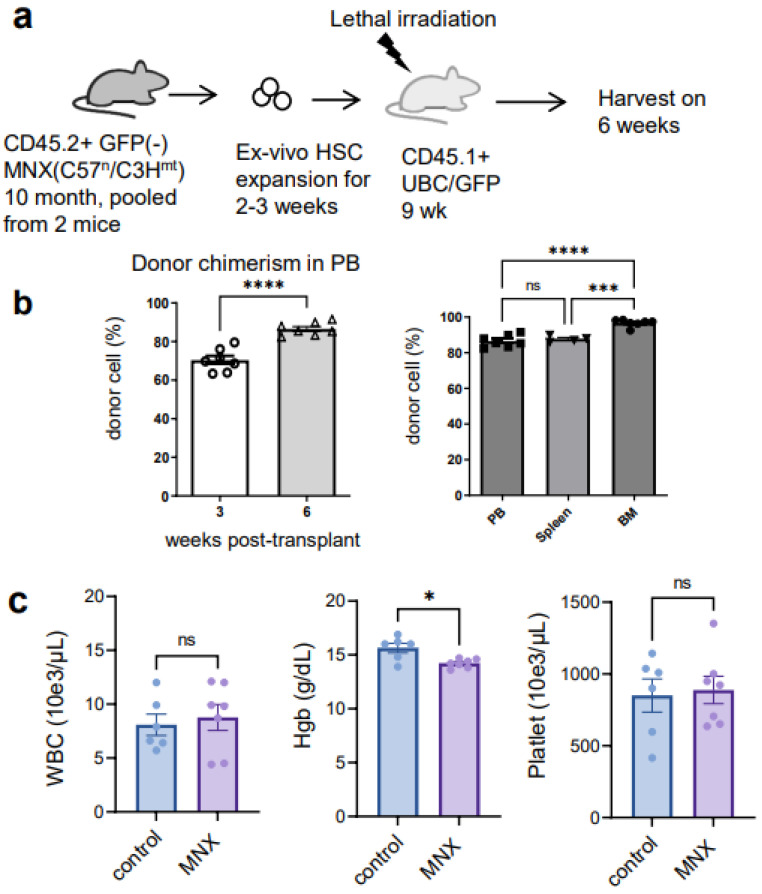
Functional assay of ex vivo expanded HSCs from aged MNX mice utilizing transplant model. (**a**) Schematic of the MNX experiment. CD150+CD34−KSL HSCs purified from CD45.2+GFP(−) MNX mice were expanded for 14 days and then transplanted into lethally irradiated recipient mice without any competitors. Each recipient received 5 × 10^5^ cells. (**b**) Donor chimerism of donor cells of live white blood cells in peripheral blood, bone marrow, and spleen at 3 or 6 weeks (harvest day). (**c**) Complete blood counts at 6 weeks. Non-transplant and non-irradiated wild type mice (control) were matched age and sex with recipient (MNX). Each dot represents an individual mouse. Data are expressed as the mean ± SEM. One-way ANOVA, Tukey’s post hoc test was employed for multiple comparisons among all groups in the graph on the right in panel (**b**) Two-tailed Student’s *t*-test was applied in panel (**c**). Statistical significance was denoted by asterisks (* *p* < 0.05, *** *p* < 0.001, and **** *p* < 0.0001) or ns; not significant.

**Figure 7 cells-12-01473-f007:**
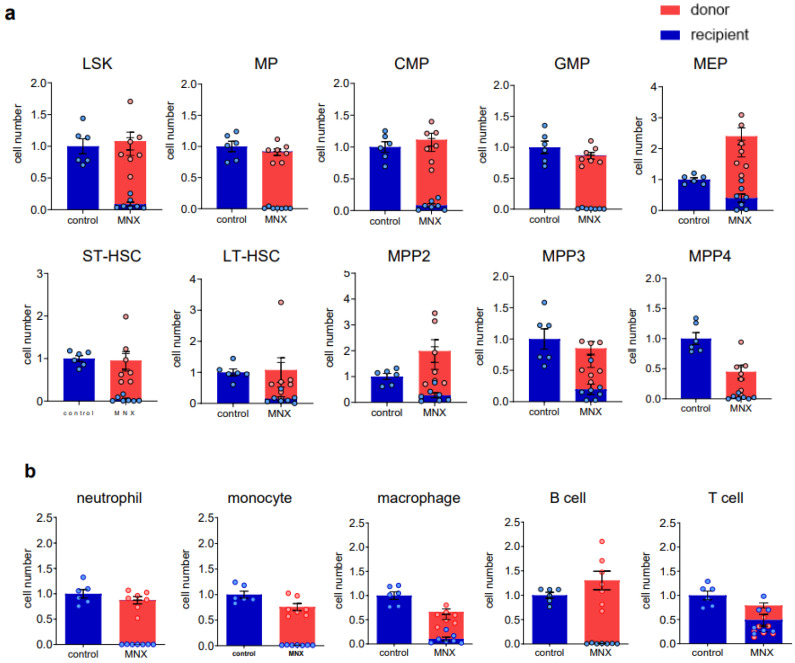
The composition of bone marrow cells in recipient mice transplanted by donor HSCs from MNX mice. Donor cells (5 × 10^5^ cells per shot) were injected in lethally irradiated recipients. (**a**) The relative cell number of donor or recipient cells in each HSPC. (**b**) The mature cell number of donor or recipient cells. Control mice were not irradiated and matched with age and sex as recipient mice. Red highlighted is donor and blue highlighted is recipient cell determined with flow cytometry (data combined from two independent transplants).

**Figure 8 cells-12-01473-f008:**
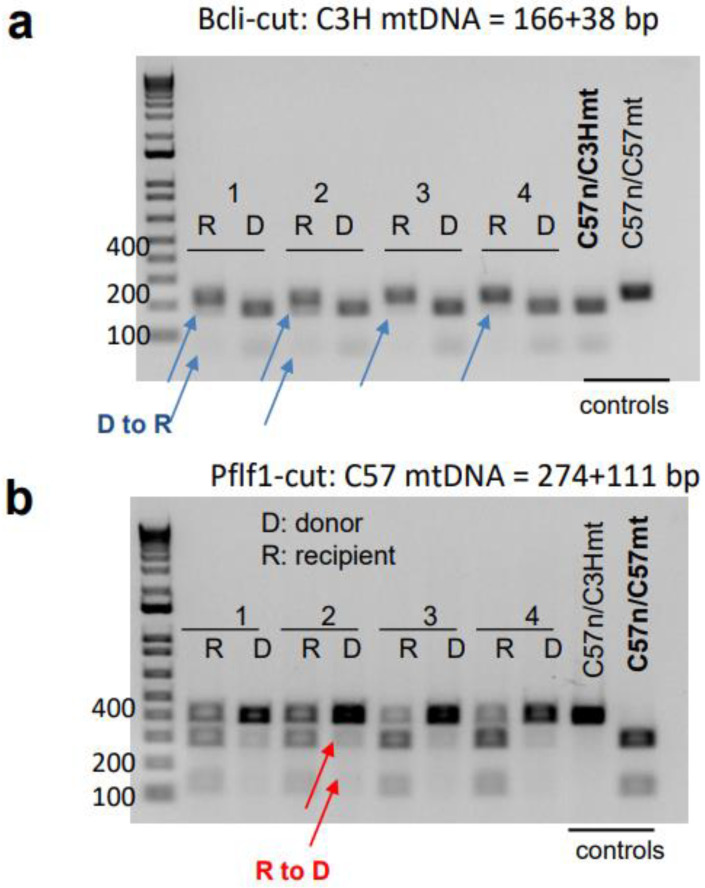
The mitochondrial DNA analysis in separated donor and recipient spleen cells. (**a**) Analysis of C3H mtDNA with Bcl1-cut in recipient (R) or donor (D) cells. D to R indicates mitochondrial transfer from donor to recipient. (**b**) C57 mtDNA evaluation with Pflf1-cut in each cell population. R to D indicates the reverse mitochondrial transfer from recipient to donor. Blue arrows indicate evidence of donor-to-recipient mitochondrial transfer. Red arrows indicate recipient-to-donor mitochondrial transfer. Each set of bands represent an individual mouse.

## Data Availability

The datasets used and/or analyzed during the current study are available from the corresponding authors on reasonable request.

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
