# Peer review of "Mitochondrial Transfer to Host Cells from Ex Vivo Expanded Donor Hematopoietic Stem Cells"

_cells, 2023, doi:10.3390/cells12111473_

Round 1

Reviewer 1 Report

N/A

Author Response

Reviewer 1

  1. The major concern is that the manuscript can be found through Google search, including the complete manuscript with figures:

“Research Square” Posted Date: April 28th, 2022

Title: Mitochondrial transfer to host cells from ex vivo expanded donor hematopoietic stem cells.

DOI: https://doi.org/10.21203/rs.3.rs-1585819/v1 https://www.researchsquare.com/article/rs-1585819/v1

Please note that the manuscript that the reviewer pointed out is a preprint in the Research Square that had not undergone a peer review and been posted on April 28th, 2022 when we previously submitted our draft to another journal, Scientific Reports, but which was not accepted. Therefore, this version is not a regular article accepted in a different journal. According to Cells editorial policies “Cells accepts submissions that have previously been made available as preprints provided that they have not undergone peer review. A preprint is a draft version of a paper made available online before submission to a journal” (https://www.mdpi.com/journal/cells/instructions#preprints).

  1. Figures presentation is a puzzle.

Fig 3. Text line 249. “Donor cell titration revealed the dose-dependent increases of donor engraftment.” Graphs, in form as presented, demonstrated that with an increasing number of donor cells, more donor cells are engrafted. What was actually expected? In this case, why does decreasing the number of recipient cells correlates with increasing donor cells? If the irradiation dose was the same for all groups, how many recipient cells survived after radiation? Or is Fig 3A, not an absolute number but the ratio of donor and recipient cells from 105 cells?

As a previous report showed (ref. Zijlmans JM, et al. PNAS, 1998), the engraftment efficiency was dependent on the number of donor hematopoietic stem cells. Also, recipient mice need to be irradiated to acquire the engraftment of donor cells via its myelosuppressive and immunosuppressive effects. Thus, the finding of Fig.3 was what we expected. Thus, we added the description regarding this in the new manuscript (page 4, line 196–197). All the groups were lethally and consistently irradiated, which means the dose to cause death of 50 % of an exposed population within 30 days (LD50/30) due to a bone marrow failure. However, no mice died in this study because we observed that donor cells were engrafted in each group although the degree of engraftment varied depending on the number of donor cells. In Fig 3A, we have shown absolute numbers of each hematopoietic cell population with a unit of 105 cells obtained from each recipient mouse.

Fig 6C. Need to revise the Description of Figure legends. Also, one-way ANOVA

As the reviewer suggested, we revised the description of Figure Legends including one-way ANOVA, describing “One-way ANOVA, Tukey’s post hoc test was employed for multiple comparisons among all groups.” (page 11, line 310)

Figure 7. The composition of bone marrow cells in recipient mice. Figure legend describes the absolute number of donor or recipient cells in each HSPS. The question: if this is an absolute cell number, why is always 1?

In Figure 7, we showed the relative cell number of each cell population in recipient mice compared to control mice, not absolute cell number. This is why the average number of control can theoretically become 1.

3. FACS is good methods but with limitations. IHC staining of bone marrow with different biomarkers can highlight more information and confirm FACS data. Line 347 and 348. Same. microscope imaging will be a good addition.

I agree with the reviewer that FACS is an appropriate tool to evaluate rare populations in bone marrow despite some limitations like lacking the structural information per se, which can be provided by IHC staining. However, IHC staining of BM cells have not technically been established as opposed to FACS, a well-accepted method in the research of hematology. Moreover, our purpose was to quantify the engraftment of donor cells and to sort out the target cells utilizing ex vivo HSC expansion as well as the panels we previously developed. Thus, we chose the flow cytometric analyses in this study.

4. A mitochondrial function assay also will be a worthy addition."

We have chosen to engraft cells that were shown before to have improved mitochondrial efficiency (from C57n;C3Hmt mitochondrio-nuclear exchange mice) when compared to C57BL6 hosts [Ref 11]. A mitochondrial function assay would be very informative, however, there is no marker present that would prospectively identify cells with mitochondrial transfer. The only way to detect a transfer is to isolate DNA which requires cell lysis. We still consider our findings significant because it has proven that the transfer using a common procedure of bone marrow grafting is possible. In our future work, we will try to engraft cells that have fluorescently labeled mitochondria, e.g. expressing mitochondria-targeted GFP.

Reviewer 2 Report

The manuscript entitled “Mitochondrial transfer to host cells from ex vivo expanded do-2 nor hematopoietic stem cells” propose the use ex vivo culture of mouse hematopoietic stem cells for expanding these cells to achieve the mitochondrial transfer from donor to host in the transplant setting. This is an interesting paper, although some of the claims he proposes about the therapeutic possibilities of this procedure are speculative and require further experimental evidence.

There are some major considerations that need to be corrected:

- It is incorrect to speak of cell grafts in media such as blood. It is better to refer to concentrations of the cell type that the authors want to express. For example, in the abstract (line 31) they state "We observed high engraftment of donor cells in peripheral blood". Correct these expressions in the text.

·   - Figures 2 and 3 summarize too much information. This complicates their interpretation. In addition, the graphs are too small and the letters and numbers on the axes cannot be read and appear blurred (Figures 2D and 2D). Fewer results per figure can be included, only those that show significant results. In Figure 2E the variable studied should be put in the name of the Y-axis and not as the title of the graph.

·      - I would suggest a list of abbreviations for better understanding.

Minor considerations:

·       - Font (size and bold) should be uniform.

·      -  Line 45 change “cancers[3] [4, 5]” by “cancers [3-5].”

·      - Line 218: please elimate “Figure 1”. It is already indicated in the legend. The same in the rest of the figures

·    - Some errors in the bibliographical references should be corrected and adapted to the style of the journal.

Author Response

There are some major considerations that need to be corrected:

- It is incorrect to speak of cell grafts in media such as blood. It is better to refer to concentrations of the cell type that the authors want to express. For example, in the abstract (line 31) they state "We observed high engraftment of donor cells in peripheral blood". Correct these expressions in the text.

As the reviewer suggested, we revised the description “high engraftment of donor cells in peripheral blood” in the abstract (page 1, line 37–38) as well as another part (page 4, line 185) in the new manuscript.

  •  - Figures 2 and 3 summarize too much information. This complicates their interpretation. In addition, the graphs are too small and the letters and numbers on the axes cannot be read and appear blurred (Figures 2D and 2D). Fewer results per figure can be included, only those that show significant results. In Figure 2E the variable studied should be put in the name of the Y-axis and not as the title of the graph.

As the reviewer suggested, we improved the size of each graph as well as the letters and numbers (Figure 2 and Figure 3) (page 7, line 254 and page 8, line 255. Page 9 and page 10, line 274-275). We removed the data of donor chimerism in PB from Fig. 2D as well as lymphocyte data from Fig. 2E. In Figure 3a, we removed the data of MEP, MPP2, MPP3, and MPP4. In Figure 3b, we removed the data of mono, mac, CD206+ Mac, CD4+ T cell, and CD8+ T cell to improve the configuration.

  •     - I would suggest a list of abbreviations for better understanding.

      As the reviewer recommended, we added the list of abbreviations as Supplementary table S2 (page 15, line 348).

Minor considerations:

  •      - Font (size and bold) should be uniform.

       As the reviewer suggested, we have revised Font being consistent with a font size of 10 in the new manuscript. The bold font was removed from each figure legend.

  •     -  Line 45 change “cancers[3] [4, 5]” by “cancers [3-5].”

      We followed this suggestion (page 2, line 50).

  •     - Line 218: please elimate “Figure 1”. It is already indicated in the legend. The same in the rest of the figures

      We followed this suggestion and remoted the unnecessary Figure title of each figure (Figure1 in page, Figure 2 in page 7 and page 8, line 255.).

  •   - Some errors in the bibliographical references should be corrected and adapted to the style of the journal.

      We included the necessary information including the full title of the paper, page range or article number, and digital object identifier (DOI) in each reference according to the style of the journal. In addition to this, we noticed some errors of citation (ref#27 and ref#43), and fixed them, as well.

Reviewer 3 Report

The paper by Hiroki Kawano and coworkers showed utility of ex vivo expanded HSC to achieve the mitochondrial transfer from donor to host in the transplant setting. Authors came to their conclusions by performing their study on in vivo model which increases the possibility of clinical use of the presented results. The paper is well organized and the results add important information to the field. However I have to point out on some editorial errors that have occurred in this work. 

Minor comments:

1. Page numbering is not correct.

2. The current structure of the publication is asymmetrical and therefore unacceptable - there is too much margin on the left side of each page.

3. Line 45, there is: "chronic inflammation, space flight and cancers[3] [4, 5]." In my opinion there should be as follows: "chronic inflammation, space flight and cancers [3-5]."

4. Line 77, there is: "diabetes [18]or". In my opinion there should be as follows: "diabetes [18] or ".

5. The approval number of the appropriate bioethics committee for animal experiments was not provided.

6. I suggest bolding only the titles of figures, and leave the rest of the legends in normal font.

7. Please reduce the spacing in legends to figures.

8. Please uniform the therm: "et al" to the correct form: "et al."

9.  Line 445, there is unnecessary space (gap) between the text.

10. The list of references is repeated twice. I suggest not using bold in the references list.

Minor editing of English language required.

Author Response

Reviewer 3

Minor comments:

  1. Page numbering is not correct.

      We tried to fix the page numbering on the provided template but were not able to change the page number in the header. Thus, we added each page number in the left bottom of each page.

  1. The current structure of the publication is asymmetrical and therefore unacceptable - there is too much margin on the left side of each page.

We changed the page structure as well as the format of each figure so that we do not make too much space on the left side of each page.

  1. Line 45, there is: "chronic inflammation, space flight and cancers[3] [4, 5]." In my opinion there should be as follows: "chronic inflammation, space flight and cancers [3-5]."

As suggested also by another reviewer, we changed the format for these references (page 2, line 50).

  1. Line 77, there is: "diabetes [18]or". In my opinion there should be as follows: "diabetes [18] or ".

We agree with you, and changed this (page 2, line 75).

  1. The approval number of the appropriate bioethics committee for animal experiments was not provided.

We added the approval code and approval date for our animal experiments (Page 2, line 95 and page 16, line 425).

  1. I suggest bolding only the titles of figures, and leave the rest of the legends in normal font.

We fixed the bold font of each figure legend according to your suggestion.

  1. Please reduce the spacing in legends to figures.

We reduced the left space in figure legends by changing the configuration of some figures (Figure 1 in page 6, Figure 3 in page 9 and 10, Figure 4 in page 11, Figure 7 in page 16 and 17).

  1. Please uniform the therm: "et al"to the correct form: "et al."

We uniformed the term in our ne manuscript according to your suggestion.

  1. Line 445, there is unnecessary space (gap) between the text.

We removed an unnecessary space in the new manuscript (page 4, line 175, page 20 line 361 and 364, and page 21, line 396).

  1. The list of references is repeated twice. I suggest not using bold in the references list.

Thank you for pointing this out. I removed one of them and avoided to use bold font in References.

Round 2

Reviewer 1 Report

Figure 3.

Remove "absolute", add exactly what you measure. same in the text.  Are Charts  format 2D stacked  column ? Please describe. 

Figure 6. One-Way ANOVA only for Fig6B right, everything else T-Test ? please clarify.

Figure 7.  Fig legend, please remove word absolute. 

Measurement of Mitochondrial function. Will be nice if possible.  Comparing mitochondrial function in recipient and donor cells and "chimera" mitochondria.  

Author Response

Figure 3.

Remove "absolute", add exactly what you measure. same in the text.  Are Charts  format 2D stacked  column ? Please describe. 

 We appreciate your suggestion. We removed “absolute” in Figure 3 and described it as “cell number” (page 8, line 282). As the reviewer pointed out, this graph is created with a format of 2D stacked column. Thus, we added a description, “The values were displayed in a 2D stacked column graph.” in the figure legend (page8, line 284-285).

Figure 6. One-Way ANOVA only for Fig6B right, everything else T-Test ? please clarify.

The reviewer is correct, and we clarified this in Figure 6 (page 11, line 316).

Figure 7.  Fig legend, please remove word absolute. 

We removed the word “absolute” and precisely added what we measured (page 12, line323).

Measurement of Mitochondrial function. Will be nice if possible.  Comparing mitochondrial function in recipient and donor cells and "chimera" mitochondria.  

As we have indicated in the previous rebuttal, identifying host cells that underwent mitochondrial transfer for functional analysis is currently impossible because detection of the transfer is by genotyping which involves cell lysis. However, Dr. Ballinger, our co-author and collaborator, has published mitochondrial functional analysis of the wild type (WT) C57BL6 vs C57n;C3Hmt mice that received C3H mitochondria [Ref. 25 and 26]. This is similar scenario to the transfer from our donor cells to the host cell. In those studies, Dr. Ballinger’s group detected that when compared to WT C57 cells, C57 cells with transferred C3H mtDNA had lower respiration and ROS production and higher resistance to stress. We have described these results in the Introduction (page 2, line 88-89)

Reviewer 2 Report

The authors have satisfactory dealt with all previous concerns. The changes are especially appreciated.

Author Response

Thank you for your objective comments

Reviewer 3 Report

Authors have responded to all my comments and made corrections in the manuscript according to my suggestion. Additionally I suggest to include abbreviations as part of the text with the headline "Abbreviations" in the manuscript but not in the form of table and not in the supplementary files. Moreover, please remove Table S2 from the main manuscript to supplementary files. I do not see other objections against accepting this paper to publication after concerning these small corrections.

Author Response

Authors have responded to all my comments and made corrections in the manuscript according to my suggestion. Additionally I suggest to include abbreviations as part of the text with the headline "Abbreviations" in the manuscript but not in the form of table and not in the supplementary files. Moreover, please remove Table S2 from the main manuscript to supplementary files. I do not see other objections against accepting this paper to publication after concerning these small corrections.

As suggested by the reviewer, we included the section of “Abbreviations” and corrected some errors (namely, the duplication of CMP and a typo of “progenitors”) in the manuscript (page16 , line 422-426), and removed Table S2 from the manuscript (page 15, line 353-354). For consistency, we also removed Supplementary Table S1 and moved it to Supplementary information